**Subject Category:**
Biology (whole organism)

behaviour/evolution

*Corydoras*, tactile interaction, group dynamics, sociality

**Author for correspondence:**
Riva J. Riley
e-mail: rjriley@post.harvard.edu

# Coping with strangers: how familiarity and active interactions shape group coordination in *Corydoras aeneus*

Riva J. Riley[1], Elizabeth R. Gillie[1,2], Cat Horswill[1,3], Rufus A. Johnstone[1], Neeltje J. Boogert[1,2] and Andrea Manica[1]

[1]Department of Zoology, University of Cambridge, CB2 3EJ Cambridge, UK
[2]Biosciences, University of Exeter, Stocker Road, Exeter EX4 4QD, UK
[3]Institute of Biodiversity, Animal Health and Comparative Medicine, University of Glasgow, Glasgow, G12 8QQ, UK

 RJR, 0000-0001-5708-7424; NJB, 0000-0002-1337-4365

Social groups composed of familiar individuals exhibit better coordination than unfamiliar groups; however, the ways familiarity contributes to coordination are poorly understood. Prior social experience probably allows individuals to learn the tendencies of familiar group-mates and respond accordingly. Without prior experience, individuals would benefit from strategies for enhancing coordination with unfamiliar others. We used a social catfish, *Corydoras aeneus*, that uses discrete, observable tactile interactions to assess whether active interactions could facilitate coordination, and how their role might be mediated by familiarity. We describe this previously understudied physical interaction, 'nudges', and show it to be associated with group coordination and cohesion. Furthermore, we investigated nudging and coordination in familiar/unfamiliar pairs. In all pairs, we found that nudging rates were higher during coordinated movements than when fish were together but not coordinating. We observed no familiarity-based difference in coordination or cohesion. Instead, unfamiliar pairs exhibited significantly higher nudging rates, suggesting that unfamiliar pairs may be able to compensate for unfamiliarity through increased nudging. By contrast, familiar individuals coordinated with comparatively little nudging. Second, we analysed nudging and cohesion within triplets of two familiar and one unfamiliar individual (where familiar individuals had a choice of partner). Although all individuals nudged at similar rates, the unfamiliar

group-mate was less cohesive than its familiar group-mates and spent more time alone. Unfamiliar individuals that nudged their group-mates more frequently exhibited higher cohesion, indicating that nudging may facilitate cohesion for the unfamiliar group-mate. Overall, our results suggest that nudges can mitigate unfamiliarity, but that their usage is reduced in the case of familiar individuals, implying a cost is associated with the behaviour.

## 1. Introduction

Animals can gain great benefits from living in groups [1,2]. Through group coordination, individuals can increase the likelihood of evading predators and improve foraging success [3,4]. Familiarity, defined as previous experience with a given other individual, has been shown to increase coordination in a variety of taxa, including birds [5] and schooling fishes [6]. For example, great tits show increased anti-predator defences within groups based on familiarity, with previous experience of nest-site neighbours leading to a greater likelihood of a familiar neighbour joining in to defend a conspecific's nest [7]. In fathead minnows, familiar groups exhibited greater shoal cohesion and more effective anti-predator behaviours (i.e. predator inspection) in the face of predation threats when compared with unfamiliar groups [3]. Likewise, familiar groups of juvenile trout responded significantly faster than unfamiliar groups to a predator attack and were more successful foragers [4].

Given the benefits of grouping with familiar individuals, it is not surprising that individuals tend to associate preferentially with familiar over unfamiliar individuals in a number of species, including cowbirds [8], lemon sharks [9] and guppies [10]. For example, female cowbirds preferentially associate with familiar group-mates when put into a group with familiar and unfamiliar conspecifics [8]. In three-spined sticklebacks (*Gasterosteus aculeatus*), the preference for familiar group-mates is stronger than the preference for group size, such that individuals tend to prefer smaller groups of familiar individuals to large groups of unfamiliar individuals [11].

Despite familiarity's many benefits, the mechanisms by which familiarity improves coordination have rarely been investigated. It seems likely that familiar individuals are better informed about each other's preferences and characteristics, and thus respond more promptly or appropriately to a partner's actions. However, quantifying such responses is challenging, as it is often unclear whether individuals are actively coordinating or not [12]. In many systems, a common way that coordination is assessed is through leader and follower roles, which represent unambiguous examples of coordination between individuals, and previous studies suggest that leader and follower roles will evolve when the cost of failing to coordinate behaviours is high [13,14]. Consequently, leadership and followership dynamics are an easily assessed measure of coordination.

In this paper, we study movement coordination within groups of bronze cory catfish (*Corydoras aeneus*). This highly social neotropical species exhibits an unusual behaviour during coordinated activities whereby individuals often nudge each other. We first noticed this behaviour in wild fish (Riley 2011, personal observation) and have since observed that this nudging behaviour occurs during both foraging and group flight responses to potential threats, in which nudging improves coordination ([15]; for a demonstration of this behaviour, see the example video provided in the electronic supplementary material). Individuals also use nudges when initiating and participating in group movements. Thus, this behaviour provides an easily scored index that might affect coordination in a variety of social contexts.

We started by demonstrating the relationship between nudging and coordinated movements. Specifically, we tested whether pairs' nudging rates were higher during coordinated movements than when partners were close together but not engaged in coordinated movements. We also investigated whether the amount of time an individual spent as the 'front fish' in a coordinated movement was related to the rate at which they nudged their partner. We then investigated the relationship between familiarity and coordination in pairs, and how this relationship was associated with nudging. We predicted that familiar individuals should be better able to coordinate compared with unfamiliar ones, and that, if nudges did facilitate coordination, we should see an increase in nudging when unfamiliar fish interacted. We then performed similar observations in triplets composed of two familiar and one unfamiliar individuals, with the expectation that the familiar individuals should preferentially interact and coordinate better, and that the unfamiliar fish should use higher level of nudging to compensate for its lack of previous experience.

# 2. Methods

## 2.1. Study species

*Corydoras* is a genus of highly social neotropical catfish widely used in the aquarium trade. In captivity, they have lifespans from 10 to 15 years [16], but their life histories in the wild are not fully known. *Corydoras* are generally benthic fish that prefer slow moving, shallow water; they are known for their marked sociality and shoaling behaviour (Nijssen in [16]). In the wild, *Corydoras aeneus*, bronze cory catfish, are social foragers that live in mixed groups of males, females and juveniles [17]. Bronze cory catfish have a slight sexual dimorphism, with females being larger and thicker-bodied than males [18]. Behaviourally, males and females are not known to differ outside of spawning and this species has never been known to engage in aggressive behaviours, not even between males during courtship and spawning [18]. We have observed that captive-bred individuals exhibit an unusual tactile interaction behaviour during coordinated activities. Wild fish were also observed using this behaviour in several small streams in the Madre de Dios locality of the Peruvian Amazon (Riley 2011, personal observation), and we are confident that this is a social interaction behaviour and not incidental interactions. Like all fishes, bronze cory catfish have lateral lines that are used to detect obstacles and sense water movement and pressure [19]; when lateral line function was disabled in a related *Corydoras* species, *Corydoras trilineatus*, obvious effects on swimming behaviour were observed, indicating that lateral line perception is important for navigation [20]. Fishes that do not interact tactilely are not known to bump into one another, even during large, dynamic shoaling movements [21], and even blind fish can shoal without colliding into one another [22]; furthermore, models of shoaling movements require repulsion forces between individuals to accurately represent shoal movement patterns [23]. None of the bronze cory catfish in this study (or in any of our previous studies) collided with objects in their enclosures or the walls of their tanks, even when placed in novel environments (as were the fish in this study).

## 2.2. Social housing husbandry

We obtained bronze cory catfish from three pet shops in Cambridgeshire: Maidenhead Aquatics Cambridge, Pet Paks Ltd, and Ely Aquatics and Reptiles. The fish used in both experiments were at least 24 weeks of age, and had been housed in the laboratory for at least six weeks prior to the start of experiments; we used individuals from this same stock population in both experiments. We maintained the fish on reverse osmosis (RO) water purified to 15 or less total dissolved solids (TDS) and re-mineralized to 105–110 ppm TDS using a commercially prepared RO re-mineralizing mix (Tropic Marin Re-mineral Tropic). The fish lived on a 12 L/12 D cycle at a temperature of 23 ± 1°C. Prior to the start of the experiment, we housed the fish in mixed-sex social housing tanks (60 × 30 × 34 cm) of 6–10 fish. The tanks were equipped with four Interpet Mini internal filters and an air stone. We fed the fish daily on a varied diet of alternating Hikari wafers (Hikari brand, USA), Tetra Prima granules (Tetra brand, Germany) and thawed frozen bloodworms (SuperFish, UK). The group composition of social housing tanks was stable for at least six weeks prior to experiment, and unfamiliar fish had not been exposed to each other for at least six months prior to the experiment, if at all. At the conclusion of each experiment, all fish were returned to the social housing tanks.

## 2.3. Pair study experimental procedure

We investigated the behaviour of familiar and unfamiliar pairs of fish; these trials were completed in three batches. Each batch consisted of four to six same-sex familiar and unfamiliar pairs to avoid spawning interactions. We analysed 26 pairs (15 familiar and 11 unfamiliar) for a total of 52 individuals. Experimental batches were tested in October 2016, November 2016 and February 2017 to allow new fish to habituate to the laboratory environment. We formed experimental pairs by randomly assigning individuals to 'familiar' or 'unfamiliar' treatments. Individuals in the 'familiar' condition were paired with an individual from their same social housing tank; unfamiliar individuals were paired with an individual from a different social housing tank (i.e. had not been exposed to each other for at least six months, if ever). Fish were not fed the morning prior to trials to encourage exploratory movement of the tank in search of food.

Pairs were placed into one of two filming tanks (45.5 × 25 × 21 cm) with a sand substrate. We filmed in both tanks simultaneously. Each filming tank had two small plastic plants in one corner of the tank to

provide cover. During each session, one filming tank contained a familiar pair while the other contained an unfamiliar pair. Familiar and unfamiliar pairs were assigned to a filming tank randomly. We filmed each pair from above with a Toshiba Camileo x100 video camera for 1 h.

## 2.4. Triplet study experiment procedure

We investigated the behaviour of familiar and unfamiliar individuals in triplets over three weeks in May–June 2017. We analysed 19 triplets for a total of 57 individuals. Each triplet consisted of two familiar individuals taken from the same social housing tank and an unfamiliar individual taken from a different tank. Triplets were composed of same-sex individuals to avoid courtship interactions, and fish were not fed prior to the trial. We placed each triplet in one of two testing arenas (47 × 30 × 29 cm) with a sand substrate. Each arena was constructed by partitioning a larger tank with a fitted opaque plexiglas sheet. We filmed in both tanks simultaneously. We used a GOPRO HERO 3 camera to film each triplet from above for 30 min.

## 2.5. Video scoring

Pair and triplet videos were scored using the same criteria, except where noted. Video scoring commenced at the first joint trip, which we defined to be a directional group movement (in which each fish was within two body lengths of another individual, and all members of the triplet or pair were moving in the same direction) lasting at least 5 s. In pairs and triplets, we were able to individually recognize fish by comparing them with their group-mates in size (i.e. body length and width) and coloration. We noted the characteristics of each individual so that we could distinguish both partners in pairs and all three individuals in triplets (i.e. the unfamiliar fish and each familiar fish). Videos were scored blind by ERG and RJR without knowing if any given pair were familiar or unfamiliar. In total, 40% of videos (randomly determined) were scored by both scorers to ensure consistency (for details, see appendix A). The remaining videos were scored by one scorer. We scored pair videos for 10 min and triplet videos for 5 min following the first joint trip. Given the added complexity of tracking interactions among triplets, scoring a shorter time compared with pairs was more practical and led to more similar numbers of interactions per individual among the two group sizes.

We quantified cohesion by estimating the amount of time two individuals spent in close proximity to one another, defined as within 7 cm (roughly two body lengths of an average sized fish). We also recorded 'nudges' (tactile interactions), which include any time fish touch one another while both are in motion or while the receiver is at rest. We are confident that, on the whole, these nudges represented directed interactions from the initiator to the recipient. While it cannot be excluded with absolute certainty that some tactile interactions were the result of unintentional collisions between individuals, we have also shown that during flight responses, individuals who perceive a threat earlier are more likely to initiate nudges with group-mates [15]. These findings add support to our interpretation that nudging is a directed interaction between individuals (as demonstrated in our electronic supplement video). Consequently, we are confident in our assessment of nudging as a discrete social interaction that individuals use in a variety of contexts, with the current study focusing on its role in the context of familiarity and group coordination.

For each nudge, we identified the actor and the recipient of the nudge and noted with which part of the body (the front-facing portion of the body, i.e. the head, or the tail) the initiator made contact with the receiver. We identified the initiator as the individual whose movement resulted in the nudge and receiver as the individual that was touched by the movement of the receiver. We also noted the region of the body that the initiator made contact with (the head, side or tail). We present analysis performed on nudges that are initiated with the front part of the initiator's body, as nudges initiated with the head will be the result of directed movement and are less likely to be incidental (i.e. a direct consequence of proximity and movement). All results are consistent and qualitatively unchanged when reanalysed with all nudges included. This analysis can be found in appendix A. Since interactions can only occur when fish are in proximity to one other, we focus on the rate of tactile interactions delivered by one fish to another while in proximity (i.e. within 7 cm). We define this 'nudging rate' as the number of nudges initiated by an individual divided by the number of seconds the pair spend in proximity (i.e. time together).

For pairs, we classified which individual was in front during coordinated movement, defined as both fish moving in the same direction with one individual at least one half of a body length in front of the other for at least three seconds. Measures of leadership and followership are commonly used to assess the extent and quality of coordination within pairs and groups [12,24–27], and movements in which there is

an apparent leader and follower represent clear examples of coordination. For these reasons, we focused on time spent in coordinated movements as a measure of active coordination (as opposed to pair cohesion, which represents a prerequisite state for pair coordination but which is not necessarily active coordination) in pairs. In familiar and unfamiliar pairs, we investigated whether coordinated movements were associated with a higher nudging rate when compared with periods when fish were in proximity to one another but not engaged in coordinated movements. If fish engaged in nudging more during coordinated movements compared with un-coordinated cohesion, then this would suggest a role of nudging in coordination (e.g. by improving information exchange among individuals).

## 2.6. Statistical analysis

All data analysis was conducted in program R (v. 3.2.2, [28]). We used a paired Wilcoxon signed-rank test in order to compare each pair's nudging rate while engaged in coordinated movements and when close to one another but not engaged in a coordinated movement. We used a Spearman rank correlation test to assess if there was an association between an individual's nudging rate and the proportion of time that individual spent in front during a coordinated movement.

We compared the proportion of time spent in coordinated movements in pairs of familiar and unfamiliar fish, and the proportion of time spent together (proportion of time together, arcsin(sqrt) transformed) using two-sample $t$-tests; transformed data were evaluated visually, and all results are qualitatively unchanged when non-parametric tests are used on the untransformed data (see appendix A). We then compared the nudging rate of familiar and unfamiliar pairs using a two-sample $t$-test.

For triplets, we used a paired $t$-test to assess differences between one randomly chosen familiar group member and the unfamiliar fish in the proportion of total time they spent in proximity to one another (we defined proportion of time together as time together/total time, which was arcsin(sqrt) transformed; transformed data were evaluated visually, and results are qualitatively unchanged when non-parametric tests are used on the untransformed data (see appendix A)). The familiar group member was chosen randomly because familiar individuals could confound one another's behaviour. Our results remain qualitatively unchanged when both familiar group-mates are included in the analysis; see appendix A for further details. We also used a paired $t$-test to assess the relationship between one randomly chosen familiar fish and the rate at which it nudged its familiar and unfamiliar group-mates. We used a one-inflated beta regression fitted using the 'gamlss' package (program R) to assess the relationship between nudging and cohesion in unfamiliar group members; the proportion of time an unfamiliar individual spent with its group-mates was the response variable, and the number of nudges the individual initiated was the explanatory variable.

# 3. Results

## 3.1. Nudge initiation patterns

Overall, in both pairs and triplets the majority of nudges were delivered with the front part of the initiator's body (i.e. the head). In pairs, 84.9% of nudges were initiated through contact from the initiator's head, and 15.1% of total nudges were initiated by the initiator's tail and therefore removed from the analysis. The proportion of nudges initiated by tail contact did not significantly differ in familiar and unfamiliar pairs (two-sample $t$-test, $t_{24} = -0.33$, $p = 0.742$). In triplets, 168 out of 1223 (13.7%) nudges were initiated by the initiator's tail. The proportion of nudges initiated by tail contact was 12.6% for both familiar individuals and 16.6% for unfamiliar individuals.

In addition, 47.3% of coordinated movements began with a nudge (i.e. a nudge occurred 0–2 s before the onset of the coordinated movements). Of the coordinated movements that began with a nudge, 47.1% began with a nudge from the front fish and 52.9% began with a nudge from the back fish. For coordinated movements that began with a nudge from the back fish, the recipient of the nudge led the coordinated movement.

## 3.2. Nudges and coordination in pairs

Nudging was associated with coordinated movements in familiar and unfamiliar pairs: nudging rates during coordinated movements were significantly higher than rates when fish were in proximity with one other but were not coordinating their movements in familiar pairs (paired Wilcoxon signed-rank

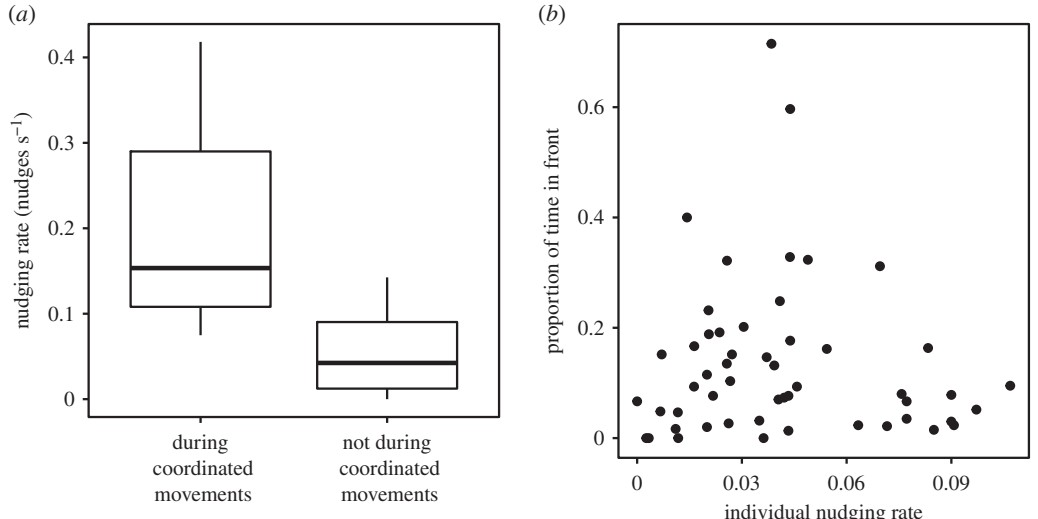

**Figure 1.** (a) Nudging rates during coordinated movements and during periods of proximity without coordinated movements in both familiar and unfamiliar pairs; (b) proportion of time in front versus rate of nudges initiated by the individual in front in familiar and unfamiliar pairs during periods of proximity. Boxplot boundaries indicate interquartile range (IQR), whiskers indicate ±1.5 IQR.

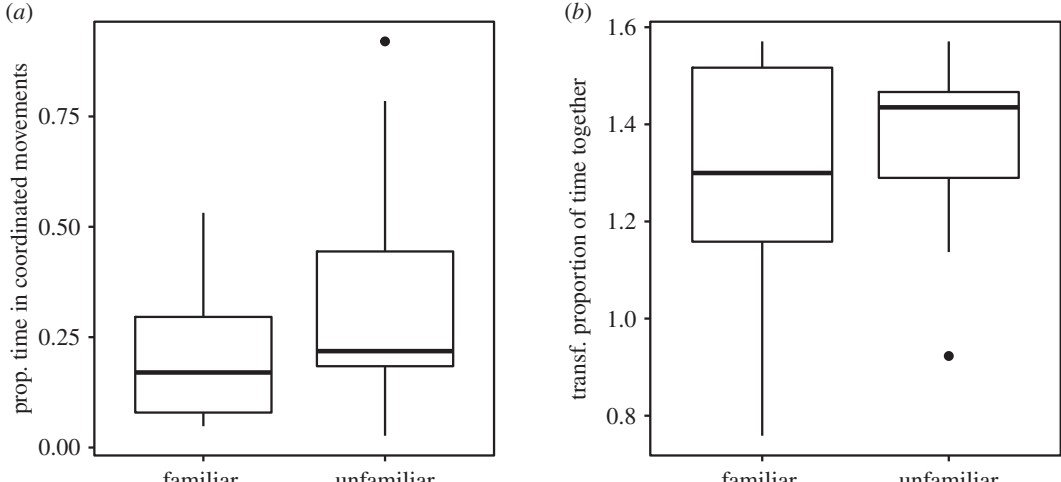

**Figure 2.** (a) Proportion of time in coordinated movements in familiar and unfamiliar groups and (b) arcsin(sqrt) transformed proportion of time together in familiar and unfamiliar groups. Boxplot boundaries indicate interquartile range, whiskers indicate ±1.5 IQR.

test, $V = 120$, $n = 15$, $p < 0.001$), unfamiliar pairs (paired Wilcoxon signed-rank test, $n = 11$, $V = 65$, $p = 0.002$) and overall (paired Wilcoxon signed-rank test, $n = 26$, $V = 350$, $p < 0.001$, figure 1a). There was no association between the amount of time an individual spent in front and its rate of initiating nudges (Spearman rank correlation, $n = 52$, $S = 22377$, $p = 0.753$, figure 1b).

## 3.3. Comparing coordination and cohesion between familiar and unfamiliar pairs

Familiarity had no effect on the level of coordination or cohesion in pairs of fish. Familiar and unfamiliar pairs spent the same proportion of time engaged in coordinated movements (two-sample $t$-test, $t_{24} = -1.83$ $p = 0.080$, figure 2a) and the same proportion of time in proximity to one another as unfamiliar pairs (two-sample $t$-test, $t_{24} = -0.85$, $p = 0.406$, figure 2b).

While patterns of coordination and nudging were similar in familiar and unfamiliar pairs, there was a significant difference in familiar and unfamiliar pairs in the rate of nudging, with individuals in unfamiliar pairs nudging each other more frequently than in familiar pairs (two-sample $t$-test: $t_{24} = -2.8$, $p = 0.010$, figure 3).

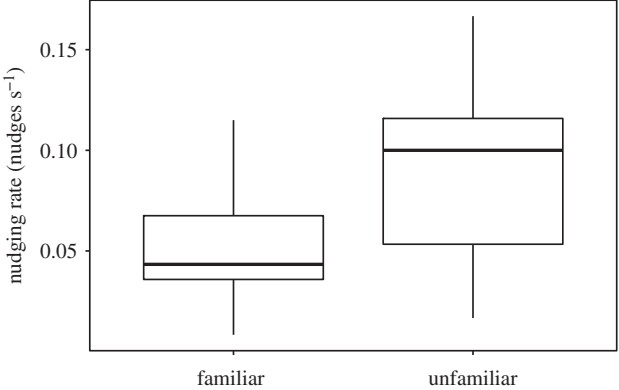

**Figure 3.** Nudging rate in pairs. Boxplot boundaries indicate interquartile range, whiskers indicate ±1.5 IQR.

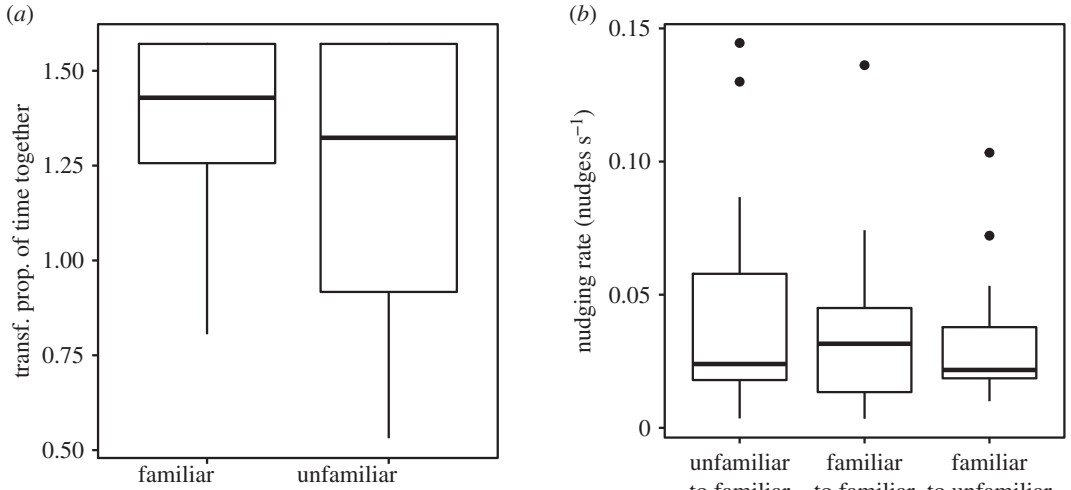

**Figure 4.** (*a*) time together (arcsin(sqrt) transformed) for one randomly selected familiar individual and the unfamiliar individual in triplets; (*b*) pairwise comparisons of nudging rates within triplets for the unfamiliar individual to one randomly selected familiar fish, as well as that familiar fish to its familiar partner and its unfamiliar partner. Boxplot boundaries indicate interquartile range, whiskers indicate ±1.5 IQR. Points beyond the whiskers are indicated.

## 3.4. Triplets

Within triplets, familiarity was associated with higher levels of cohesion. A randomly selected familiar fish spent a higher proportion of time in proximity to one other fish than its unfamiliar group-mate (paired *t*-test, $t_{18} = -2.6$, $p = 0.016$, figure 4*a*). However, when fish were in close proximity to one another, familiarity between individuals did not influence nudging rate in the familiar fish (paired *t*-test, $t_{18} = 0.8$, $p = 0.41$, figure 4*b*).

There was a significant relationship between the unfamiliar fish's total number of nudges and the proportion of time the unfamiliar fish spent with at least one group-mate (one-inflated beta regression, $t_{15} = -3.9$, $p = 0.001$, figure 5), i.e. there is a significant positive association between nudging and cohesion in unfamiliar fish, such that increased nudging is associated with increased cohesion.

# 4. Discussion

The presence of easily identifiable nudges in bronze cory catfish allows us to quantify the link between coordination and interactions among group members (even though we note that nudges are only one method of interacting). Specifically, nudging rates are significantly higher during coordinated movements than when pairs are not engaged in a coordinated movement. The fact that an individual's nudge initiation rates are not correlated with the individual's time in front suggests that nudging is not used exclusively by leaders, but by both the 'front fish' and the 'back fish' to maintain coordination

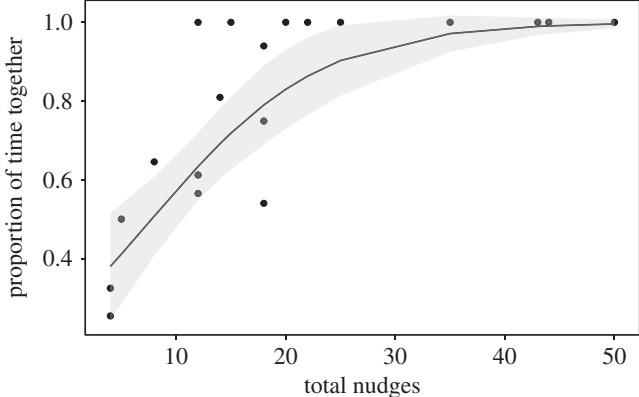

**Figure 5.** Relationship between the proportion of time together and total nudges in the unfamiliar group-mate. Each data point is plotted with the predictions of a one-inflated beta regression (solid line). The grey shaded region represents the 95% confidence interval for this relationship.

during joint movements. This suggests that nudging is a behaviour that bronze cory catfish individuals can use while coordinating with others both during group flight responses and while exploring a new area, as the pairs and triplets in this study were obliged to do. This is consistent with other systems in which communication behaviours regulate group coordination, as in mouse lemurs, which use olfactory signals to regulate inter-group spatial coordination and acoustic signals to regulate intra-group cohesion and coordination [29]. Similar examples exist in birds, for example, in green woodhoopoes vocalizations are used to maintain group cohesion while moving to new territory [30]. The use of tactile interactions by bronze cory catfish introduces interactions that use a different sensory modality than these examples, and which are strongly associated with coordinated movements directed by a front fish. Consequently, we used nudging as one metric to assess how familiarity affects coordination in triplets and pairs of bronze cory catfish.

We found that familiarity affected individuals differently based on group size. Individuals in pairs do not have a choice of group-mates, and without this choice, familiar and unfamiliar pairs spent similar amounts of time together and exhibited similar patterns of coordination and cohesion. However, it appears that unfamiliar pairs may have to engage in a significantly higher nudging rate in order to exhibit the same degree of coordination as familiar pairs. In triplets, our results were consistent with the established literature from a variety of taxa on the effect of familiarity on group coordination. Individuals in a triplet with one familiar group-mate and one unfamiliar group-mate spent more time in proximity with their familiar group-mate, although they nudged both group-mates at similar rates. The effect of cohesion on familiarity is in line with our expectations, and similar results obtained in a number of other fishes [3,10]. A deeper analysis of this pattern reveals that, although unfamiliar group members generally exhibited lower social cohesion, many of the unfamiliar group members maintained high cohesion with their familiar group-mates. The unfamiliar group members that did exhibit high cohesion also exhibited increased nudging beyond what we would expect from proximity effects; this underscores the potential importance of nudging as behaviour individuals can deploy to facilitate coordination when they are unfamiliar with their group-mates.

The higher nudging rates in unfamiliar, compared with familiar, pairs, as well as the relationship between nudging frequency and cohesion in unfamiliar group members in triplets, suggest that nudging may play a role in aiding coordination between individuals without prior experience (and therefore without social information) about one another. However, the results that familiar pairs seemed to downregulate their nudging frequency when it was no longer required, and that the unfamiliar group members in triplets were generally less socially cohesive than their familiar group-mates, imply that coordination with unfamiliar individuals is likely to carry some cost. Because cory catfish rely on camouflage to avoid predators and freeze when threatened, it seems likely that individuals might avoid excessive use of nudges, as this is likely to make them more conspicuous to predators. For familiar group members in triplets, familiar individuals had a choice of partners, and therefore maintaining social cohesion with their familiar partner required less potentially costly nudges. Because familiar group members already had a group-mate with which they could easily coordinate, they may not have exerted the same increased nudging rates as unfamiliar pairs (in which individuals did not have a choice of partner). As a result, unfamiliar individuals may have been more likely to spend more time alone, because they were left behind by their more

coordinated group-mates. The unfamiliar group member would consequently face a trade-off between nudging more and increasing cohesion with their group-mates, or remaining still and being less conspicuous.

Because unfamiliar pairs engage in coordinated movements as frequently as familiar pairs, but require more nudges, while unfamiliar individuals in triplets require more nudges to achieve similar cohesion levels as their familiar group-mates, it seems that familiarity may reduce the level of interaction necessary to achieve effective coordination. The unfamiliar group members' association between nudging and increased cohesion in triplets supports this possibility. This suggests that familiar individuals in groups may be able to achieve greater levels of coordination, because they have had more chances to interact with one another previously and can respond to one another more effectively; this is consistent with the effect of familiarity in other species, including lemon sharks [9]. Individuals can then initiate and respond to nudges more effectively based on the previous interactions they have had with their familiar group-mates.

Given the fact that the pairs in our study could achieve the same level of coordination via either familiarity or increased nudging, there may be underlying effects of previous interactions on an individual's response to conspecifics. Previous social experience can have dramatic effects on social preference, as in barnacle geese [31], which could potentially impact many important social behaviours. Evidence from guppies and deer mice suggests that familiar groups are capable of social learning at a faster rate than unfamiliar ones [32,33], which could be due to increased cohesion and individuals' (both demonstrator and follower) greater receptiveness to familiar group-mates. In addition, familiarity reduces aggression in many species, with an individual less likely to display aggressive behaviours toward an individual with which it has prior experience [6,34,35]. Future work might investigate how familiarity leads to such outcomes. Familiarity may lead to greater sensitivity to others, which in turn increases social learning potential and reduces the risks of competitive interactions.

Finally, our study shows that the relationship between familiarity and coordination is not necessarily a given but can be influenced by individual interactions between group-mates. There is substantial literature exploring the negative effects of unfamiliarity on groups, but many animals probably use tactics to mitigate these effects. Our results suggest that species that can actively coordinate with potential group-mates will selectively employ such tactics to obviate the disadvantages of unfamiliarity, but that individuals will do so only when the costs of poor coordination associated with unfamiliarity outweigh the costs of active efforts to coordinate with unfamiliar conspecifics. The costliness of unfamiliarity (likewise, the benefits of familiarity), as well as the necessity for animals to coordinate with others, are a potential selection pressure for the evolution of efficient communication systems.

Ethics. This study was approved via a non-regulated use of animals in scientific procedures application (consistent with UK's animal welfare legislation ASPA) through the University of Cambridge. It was approved through the University of Cambridge's Ethical Review Process; it was approved and presented by the Named Veterinary Surgeon and the Named Animal Care and Welfare Officer (NACWO) for the Zoology department.
Data accessibility. The data for this manuscript are available in the electronic supplementary material.
Authors' contribution. R.J.R. conceived the experiment and participated in experimental design, execution, video scoring, data analysis, manuscript writing and manuscript editing; E.R.G. contributed to experimental design, execution and video scoring; C.H. contributed to data analysis and manuscript editing; R.A.J. assisted with data analysis and manuscript editing; N.J.B. and A.M. contributed to with experimental design, data analysis and manuscript editing.
Competing interests. The authors have no conflicts of interest to declare.
Funding. R.J.R. was funded by a Herchel Smith Postgraduate Fellowship; N.J.B. was funded by a Dorothy Hodgkin Fellowship.
Acknowledgements. We wish to thank Dr François-Xavier Dechaume-Moncharmont and one anonymous reviewer for their insightful feedback on the manuscript, James Savage and Jim Allen for their advice and contributions to this project, and Arne Jungwirth for his advice throughout the project and thoughtful comments on the manuscript.

# Appendix A. Coping with strangers: how familiarity and active interactions shape group coordination in Corydoras aeneus
## A.1. Description of scoring protocol

Each tactile interaction consisted of two fish making visible physical contact in the video. We only scored interactions that occurred when both fish were on the bottom of the filming tank, because when fish were

higher in the water column it was impossible to tell if fish were actually touching or if one was merely above the other.

Each tactile interaction has an initiator and a receiver:

— An initiator is the individual whose movement resulted in the nudge.
— I.e. if two fish are in proximity at rest, and Fish I begins to swim and makes contact with Fish II as a result, Fish I has initiated the interaction.
— A receiver is the individual who was touched by the movement of the receiver.
— We noted the region of the body that the initiator made contact with and the region of the body on which the receiver was contacted.

A receiver can ignore the interaction.

— 'Ignoring' an interaction occurs when the receiver is initially at rest prior to the interaction, and does not move (beyond physical recoil from the nudge itself) within 3 s following the interaction.
— 'Ignoring' interactions were not counted in the total number of interactions, or in the individual initiations.
— There were few instances of 'ignoring' reactions in the videos.

A mutual interaction is one in which both fish are moving toward one another and make contact. Thus both 'initiate' and 'receive' the interaction.

— Mutual interactions counted toward the total number of interactions, but were not included in individual counts.

Total interaction counts include: interactions initiated by fish I + interactions initiated by fish II + mutual interactions.

Fish were defined as 'together' if they were within two body lengths (7 cm) of another fish, and apart if they were further than two body lengths from their group-mate(s).

Interaction rates are defined as: number of interactions initiated by fish I divided by the number of seconds fish I spent together with a given group-mate.

We defined 'time in front' as a measure of leadership. We only defined this measure for pairs. We defined an individual as being 'in front' if both members of the pair were swimming in the same direction and the individual was at least one half of a body length in front of its partner for at least 3 s.

We scored each instance that each individual was in front, noting start and end times, and took the sum of the number of seconds each individual spent in front.

Total time in front is defined as: (Fish I time in front) + (Fish II time in front).

## A.2. Assessment of consistency of the scoring protocol

### A.2.1. For pairs

For the first two batches of pair videos, we used a set of one to three randomly selected videos as a training set. The training set was scored by both Riva Riley and Beth Gillie, and scores were compared and inconsistencies resolved by referring to the scoring protocol and reaching a consensus about each interaction. After the training set, an additional comparison set of videos was scored by both scorers, and the scores for each measure (number of interactions initiated by each fish, mutual, number of interactions ignored, amount of time spent together, amount of time each individual spent as front fish) compared. The comparison set was also consensus scored in the process of comparison. If all measures were within 85% for both scorers, scores were deemed consistent. If any measure was not within 85% for both scores, an additional training set was completed by both scorers, followed by an additional comparison set. For the first batch of videos, we completed two training sets and two comparison sets— by the second comparison set, the two scorers were consistent by at least 85%. The rest of the videos were scored individually by either Beth Gillie or Riva Riley. For the second batch of videos, we completed one training set and one comparison set, which was found to be at least 85% consistent. The remainder of the videos from the second batch was scored individually by either Riva Riley or Beth Gillie. For the third batch, a comparison set was completed by both scorers, and was consensus scored in

the process of comparison—both scores for all videos were at least 85% consistent. The remainder of the videos for the third batch was scored individually by Riva Riley or Beth Gillie.

### A.2.2. For triplets

We used a set of one to three randomly selected videos as a training set. The training set was scored by both Riva Riley and Beth Gillie, and scores were compared and inconsistencies resolved by referring to the scoring protocol and reaching a consensus about each interaction. After the training set, an additional set of comparison videos were scored by both scorers, and each measure (number of interactions initiated by each fish, mutual, number of interactions ignored, amount of time spent together) was compared across both scorers. If all measures were within 85% for both scorers, scores were deemed consistent. The comparison set was found to be consistent by this definition, and was also consensus scored in the process of comparison. The rest of the videos were scored by Riva Riley.

### A.2.3. Other scoring notes

Only pairs and triplets consisting of same-sex individuals were scored. It was clear if opposite-sex pairs or triplet groups occurred because vigorous courtship interactions ensued. These groups (three pairs and one triplet) were excluded from analysis. Therefore, 30 pairs and 19 triplets were initially tested, for 60 individuals participating in the pair experiment and 57 individuals in the triplet experiment.

## A.3. Results with all nudges included

Note: only results involving nudges are presented.

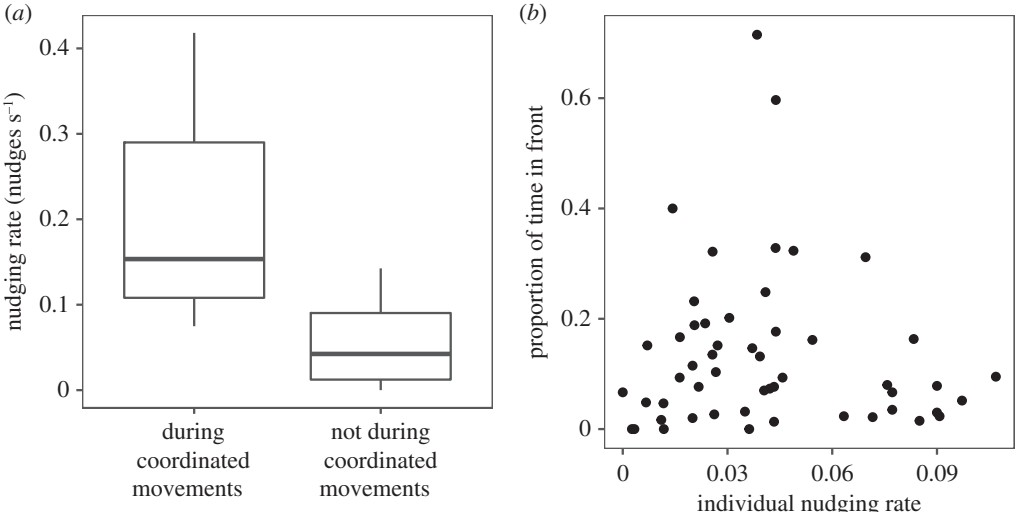

**Figure 6.** (*a*) Nudging rates during and outside of coordinated movements in unfamiliar pairs; (*b*) proportion of time in front (arcsin(sqrt) transformed) versus rate of nudges initiated by the individual in front in unfamiliar pairs. Boxplot boundaries indicate interquartile range, whiskers indicate ±1.5 IQR. Nudging was associated with coordinated movements in familiar and unfamiliar pairs: nudging rates during coordinated movements were significantly higher than rates when fish were in proximity with one other but were not coordinating their movements in familiar pairs (paired Wilcoxon signed-rank test, $V = 120$, d.f. = 14, $p < 0.001$), unfamiliar pairs (paired Wilcoxon signed-rank test, d.f. = 11, $V = 66$, $p < 0.001$) and overall (paired Wilcoxon signed-rank test, d.f. = 25, $V = 351$, $p < 0.001$, figure 2a). There was no association between the amount of time an individual spent in front and its rate of initiating nudges (Spearman rank correlation, $n = 52$, $S = 21\,796$, $p = 0.624$).

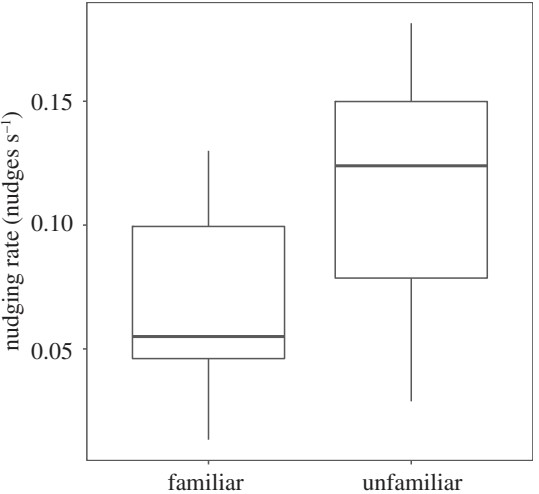

**Figure 7.** Nudging rate in familiar and unfamiliar pairs (two-sample $t$-test: $t_{24} = 2.68$, $p = 0.013$). Boxplot boundaries indicate interquartile range, whiskers indicate ±1.5 IQR.

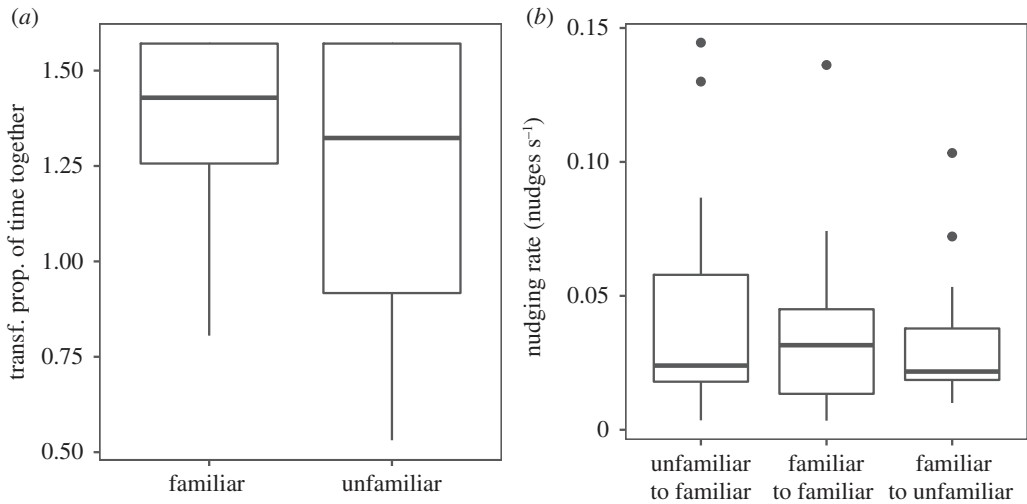

**Figure 8.** ($a$) Time together (arcsin(sqrt) transformed) for one randomly selected familiar individual and the unfamiliar individual in triplets; ($b$) pairwise comparisons of nudging rates within triplets for the unfamiliar individual to one randomly selected familiar fish, as well as that familiar fish to its familiar partner and its unfamiliar partner (paired $t$-test, $t_{18} = 1.4$, $p = 0.17$, figure 4$b$). Boxplot boundaries indicate interquartile range, whiskers indicate ±1.5 IQR. Points beyond the whiskers are indicated.

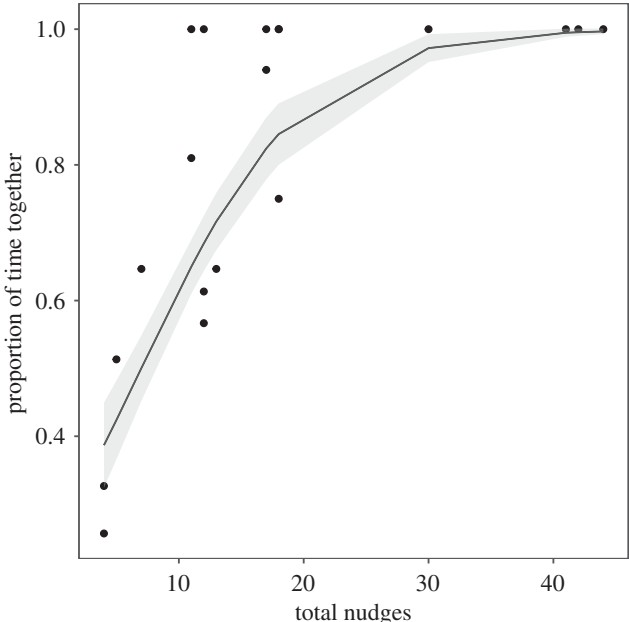

**Figure 9.** Relationship between the proportion of time together and total nudges in the unfamiliar group-mate (one-inflated beta regression, $t_{15} = -4.1$, $p < 0.001$). Each data point is plotted with the predictions of a one-inflated beta regression (solid line). The grey shaded region represents the 95% confidence interval for this relationship.

## A.4. Non-parametric equivalents for transformed data

All results were qualitatively unchanged when non-parametric tests were performed on raw (i.e. not transformed) data.

### A.4.1. Pairs

The proportion of time familiar and unfamiliar pairs spent in proximity to one another was similar (Wilcoxon signed-rank test, $W = 68.5$, $p = 0.481$).

### A.4.2. Triplets

Unfamiliar individuals in a triplet with two familiar individuals spend significantly less time in proximity to their group-mates as compared with one randomly chosen familiar individual (paired Wilcoxon signed-rank test, $V = 85$, $p = 0.006$).

## A.5. Triplet results with both familiar individuals included in analysis

### A.5.1. Triplets

Within triplets, familiarity was associated with higher levels of cohesion: the two familiar fish spent a higher proportion of time in proximity to one another than they did near the unfamiliar fish (ANOVA, $F_{1,18} = 14.1$, $p = 0.0006$, figure 4a; removing the outlier does not impact the result). However, when fish were in proximity to one another, there was no effect of familiarity on the rate at which they nudged each other (ANOVA, $F_{2,18} = 2.08$, $p = 0.131$, figure 4b).

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
