## [Reviewer comments · Royal Society Open Science]

Review History

RSOS-190587.R0 (Original submission)

Review form: Reviewer 1

Is the manuscript scientifically sound in its present form?

Yes

Are the interpretations and conclusions justified by the results?

Yes

Is the language acceptable?

Yes

Is it clear how to access all supporting data?

Yes

Do you have any ethical concerns with this paper?

No

Have you any concerns about statistical analyses in this paper?

No

Recommendation?

Accept with minor revision (please list in comments)

Comments to the Author(s)

The manuscript "Coping with strangers: how familiarity and active interactions shape group coordination in *Corydoras aeneus*" investigates whether tactile interactions facilitate coordination in pairs and triplets of familiar and unfamiliar fish. In general, I found the manuscript to be a sound and interesting piece of work on an understudied aspect of social interactions in groups (namely tactile interactions). I have relatively minor comments, however, I would like to the authors to address these before I can recommend publication.

Line 145: Please confirm whether it 26, and not 27, pairs that were analysed (to mirror the number of individuals that were tested).

Line 141 & 161: Please confirm whether the same fish were used in the pairs study as were used in the triplet study.

Line 212: Details of the appendix should be referred to that highlight how the authors defined initiator and receiver.

Line 235: Please confirm how you tested for normality following transformation of these data.

Line 251: I do not think that a quadratic function is an appropriate model to fit to the data here. There is no evidence that as the total number of nudges increases, the proportion of the time together decreases. Have the authors tried to fit a logistic regression to these data? This would satisfy the binomial nature of the data (the fish are together or not) and second this limits the predicted proportions between 0 and 1. This would be a more suitable fit than the quadratic curve, and would not affect interpretation of results.

Line 259: t_{24} - 24 should be subscript.

Line 261: It would be useful to report the % of nudges from the front fish and back fish as percentages out of the 47.3% of nudges that initiated coordinated movement. i.e. they will both be approximately 50% (and not ~25%).

Figure 1: In the text (line 275) the authors report that these data are from both familiar and unfamiliar fish pairs, whereas in the figure the authors report this is data from just the unfamiliar pairs. Please confirm.

Review form: Reviewer 2 (François-Xavier Dechaume-Moncharmont)**Is the manuscript scientifically sound in its present form?**

Yes

Are the interpretations and conclusions justified by the results?

No

Is the language acceptable?

Yes

Is it clear how to access all supporting data?

Yes

Do you have any ethical concerns with this paper?

No

Have you any concerns about statistical analyses in this paper?

No

Recommendation?

Major revision is needed (please make suggestions in comments)

Comments to the Author(s)

This manuscript is a descriptive study about a novel behaviour possibly involved in coordinated movement in a fish species. The MS is very clearly written in a direct and engaging style. I also greatly appreciated the clarity of the methodological description in the Material and Methods section and the important Appendix (would it be possible to move some of this material in the main text?). Most, but not all, conclusions of the study are well supported by the dataset. My overall impression about this manuscript is positive. Yet, some points deserve clarification prior final acceptance.

Major comments.

My main criticism is related to the absence of direct investigation of the causal relationship between nudging rate and the degree of coordination between fish. The authors only provide very indirect evidences of this relationship. Strong statements are made in the discussion: "unfamiliar pairs had to engage in a significantly higher nudging rate in order to achieve the same degree of coordination as familiar pairs" (Lines 353-354), "nudging can play a role in aiding coordination between individuals" (line 369). Yet, the authors do not really provide data about the quality or the degree of coordination between fish. To confidently claim that this nudges are involved in the coordination between fish, the authors must first address several issues:

(1) It is crucial to quantify the coordination between fish and characterize the quality of this coordination. There are many metrics of coordination in fish school to be found in the vast literature on collective movement

(2) Then, the authors must correlate the degree of coordination with the nudging rates.

Were the individually marked in order to be recognized? Firstly, it is a crucial methodological question: how did they recognized each fish (familiar vs non-familiar from a triplet of fish) in the 30-minutes video recordings? The MS is unclear on that matter. Secondly, and maybe more interestingly, ID information allows for the assessment of inter-individual differences in the initiation of these nudges. Are some fish more prone to nudge? Are their nudging rate repeatable?

Minor comments.

Lines 164 and followings. About the fish used in the second experiment (based on triplet), were they different fish from experiment 1, or fish randomly sampled from the same stock?

Lines 190-201. These sentences seem exogenous, here. They break the flow of the paragraph, which deals with the "intentionality" or the "accidental nature" of the nudges. Could it be possible to move these sentences earlier, maybe in the introduction section?

Line 252. Provide complete reference (R software foundation 2018).

Line 257 and followings. While not incorrect, the presentation of the results is confusing. Earlier in the MS (lines 213-214), the authors claimed that "[they] We present analysis performed on nudges that are initiated with the FRONT PART of the initiator's body". Yet, in this Result paragraph, they systematically presented stats about the nudges initiated by the TAIL. In addition, I do not understand how it is possible that "coordinated movements began with a nudge from the back fish." What does that mean? The fish that was touched led the collective movement?

Fig. 1. "tras" is an unusual abbreviation. In addition, the legend is not self-explicit. The reader must be informed that the nudging rate corresponds not to plain number of nudges per unit of time, but to a number of nudges per unit of time spent in proximity. This is not a minor detail because this metric is correct for the lower spatial proximity between unfamiliar fish.

Lines 320 and following. The quadratic model is a very bad modelling choice, here. By nature, the proportion of time is bounded between 0 and 1. In addition, it is unsurprising that the proportion of time spent together mechanically increases with the absolute number of nudges (and vice versa) up to a point where the number of nudges is so large that the two fish literally stick together. The number of nudges cannot decrease, as suggested by the quadratic relationship.

--

Dr François-Xavier Dechaume-Moncharmont
University of Burgundy, Dijon, France
fx.dechaume@u-bourgogne.fr

Decision letter (RSOS-190587.R0)

03-Jul-2019

Dear Ms Riley,

The editors assigned to your paper ("Coping with strangers: how familiarity and active interactions shape group coordination in *Corydoras aeneus*") have now received comments from reviewers. We would like you to revise your paper in accordance with the referee and Associate Editor suggestions which can be found below (not including confidential reports to the Editor). Please note this decision does not guarantee eventual acceptance.

Please submit a copy of your revised paper before 26-Jul-2019. Please note that the revision deadline will expire at 00.00am on this date. If we do not hear from you within this time then it will be assumed that the paper has been withdrawn. In exceptional circumstances, extensions may be possible if agreed with the Editorial Office in advance. We do not allow multiple rounds of revision so we urge you to make every effort to fully address all of the comments at this stage. If deemed necessary by the Editors, your manuscript will be sent back to one or more of the original reviewers for assessment. If the original reviewers are not available, we may invite new reviewers.

- Data accessibility

If you wish to submit your supporting data or code to Dryad (<http://datadryad.org/>), or modify your current submission to dryad, please use the following link:
<http://datadryad.org/submit?journalID=RSOS&manu=RSOS-190587>

- Competing interests

- Authors' contributions

- Acknowledgements

- Funding statement

Kind regards,

Alice Power

Editorial Coordinator

on behalf of Kevin Padian (Subject Editor)

Associate Editor's comments:

Two reviewers have provided commentary on your work, each recommending a range of changes, some that should be a quick fix, others are more substantial and will require further reviewer oversight when you submit your revised paper. Please ensure that you provide a revised manuscript (ideally identifying changes clearly with some format of tracked changes) and a point-by-point response to the comments. Note that in general, only one round of revision is permitted, so we urge you to work hard to fully address the referees' concerns.

Subject Editor Comments to Author:

Thank you for your submission. As you'll see the reviewers were generally positive but there are some issues that need to be addressed. We look forward to your revision.

Comments to Author:

Reviewers' Comments to Author:

Reviewer: 1

Comments to the Author(s)

The manuscript "Coping with strangers: how familiarity and active interactions shape group coordination in *Corydoras aeneus*" investigates whether tactile interactions facilitate coordination in pairs and triplets of familiar and unfamiliar fish. In general, I found the manuscript to be a sound and interesting piece of work on an understudied aspect of social interactions in groups (namely tactile interactions). I have relatively minor comments, however, I would like to the authors to address these before I can recommend publication.

Line 145: Please confirm whether it 26, and not 27, pairs that were analysed (to mirror the number of individuals that were tested).

Line 141 & 161: Please confirm whether the same fish were used in the pairs study as were used in the triplet study.

Line 212: Details of the appendix should be referred to that highlight how the authors defined initiator and receiver.

Line 235: Please confirm how you tested for normality following transformation of these data.

Line 251: I do not think that a quadratic function is an appropriate model to fit to the data here. There is no evidence that as the total number of nudges increases, the proportion of the time together decreases. Have the authors tried to fit a logistic regression to these data? This would satisfy the binomial nature of the data (the fish are together or not) and second this limits the predicted proportions between 0 and 1. This would be a more suitable fit than the quadratic curve, and would not affect interpretation of results.

Line 259: t_{24} - 24 should be subscript.

Line 261: It would be useful to report the % of nudges from the front fish and back fish as percentages out of the 47.3% of nudges that initiated coordinated movement. i.e. they will both be approximately 50% (and not ~25%).

Figure 1: In the text (line 275) the authors report that these data are from both familiar and unfamiliar fish pairs, whereas in the figure the authors report this is data from just the unfamiliar pairs. Please confirm.

Reviewer: 2

Comments to the Author(s)

This manuscript is a descriptive study about a novel behaviour possibly involved in coordinated movement in a fish species. The MS is very clearly written in a direct and engaging style. I also greatly appreciated the clarity of the methodological description in the Material and Methods section and the important Appendix (would it be possible to move some of this material in the main text?). Most, but not all, conclusions of the study are well supported by the dataset. My overall impression about this manuscript is positive. Yet, some points deserve clarification prior final acceptance.

Major comments.

My main criticism is related to the absence of direct investigation of the causal relationship between nudging rate and the degree of coordination between fish. The authors only provide very indirect evidences of this relationship. Strong statements are made in the discussion: "unfamiliar pairs had to engage in a significantly higher nudging rate in order to achieve the same degree of coordination as familiar pairs" (Lines 353-354), "nudging can play a role in aiding coordination between individuals" (line 369). Yet, the authors do not really provide data about the quality or the degree of coordination between fish. To confidently claim that this nudges are involved in the coordination between fish, the authors must first address several issues:

(1) It is crucial to quantify the coordination between fish and characterize the quality of this coordination. There are many metrics of coordination in fish school to be found in the vast literature on collective movement

(2) Then, the authors must correlate the degree of coordination with the nudging rates.

Were the individually marked in order to be recognized? Firstly, it is a crucial methodological question: how did they recognized each fish (familiar vs non-familiar from a triplet of fish) in the 30-minutes video recordings? The MS is unclear on that matter. Secondly, and maybe more interestingly, ID information allows for the assessment of inter-individual differences in the initiation of these nudges. Are some fish more prone to nudge? Are their nudging rate repeatable?

Minor comments.

Lines 164 and followings. About the fish used in the second experiment (based on triplet), were they different fish from experiment 1, or fish randomly sampled from the same stock?

Lines 190-201. These sentences seem exogenous, here. They break the flow of the paragraph, which deals with the "intentionality" or the "accidental nature" of the nudges. Could it be possible to move these sentence earlier, maybe in the introduction section?

Line 252. Provide complete reference (R software foundation 2018).

Line 257 and followings. While not incorrect, the presentation of the results is confusing. Earlier in the MS (lines 213-214), the authors claimed that "[they] We present analysis performed on nudges that are initiated with the FRONT PART of the initiator's body". Yet, in this Result paragraph, they systematically presented stats about the nudges initiated by the TAIL. In addition, I do not understand how is it possible that "coordinated movements began with a nudge from the back fish." What do that mean? The fish that was touched lead the collective movement?

Fig. 1. "trasf" is an unusual abbreviation. In addition, the legend is not self-explicit. The reader must be informed that the nudging rate correspond not to plain number of nudges per unit of time, but to a number of nudge per unit of time spent in proximity. This is not a minor detail because this metric correct for the lower spatial proximity between unfamiliar fish.

Lines 320 and following. The quadratic model is a very bad modelling choice, here. By nature, the proportion of time is bounded between 0 and 1. In addition, it is unsurprising that the proportion of time spend together mechanically increases with the absolute number of nudges (and vice versa) up to a point where the number of nudges is so large that the two fish literally stick together. The number of nudge cannot decrease, as suggested by the quadratic relationship.

--

Dr François-Xavier Dechaume-Moncharmont
University of Burgundy, Dijon, France
fx.dechaume@u-bourgogne.fr

Author's Response to Decision Letter for (RSOS-190587.R0)

See Appendix A.

RSOS-190587.R1 (Revision)

Review form: Reviewer 1

Is the manuscript scientifically sound in its present form?

Yes

Are the interpretations and conclusions justified by the results?

Yes

Is the language acceptable?

Yes

Do you have any ethical concerns with this paper?

No

Have you any concerns about statistical analyses in this paper?

No

Recommendation?

Accept as is

Comments to the Author(s)

The authors have addressed all my previous concerns and I am happy to recommend the manuscript for publication without further changes.

Review form: Reviewer 2 (François-Xavier Dechaume-Moncharmont)

Is the manuscript scientifically sound in its present form?

Yes

Are the interpretations and conclusions justified by the results?

Yes

Is the language acceptable?

Yes

Do you have any ethical concerns with this paper?

No

Have you any concerns about statistical analyses in this paper?

No

Recommendation?

Accept with minor revision (please list in comments)

Comments to the Author(s)

I carefully read through the new version of the MS and the point-by-point response letter. The authors have made appreciated effort to convincingly address all the comments made by the referees. The MS is now much clearer. The introduction, materials and methods, and results sections have been rewritten, rearranged and occasionally expanded. I also greatly appreciated the new thorough statistical analyses and the clarity of the methodological description. Despite the fact that I still regret the absence of information about repeatability or inter-individual consistency of the described behaviours, the conclusions of the study are well supported by the neat dataset. I now recommend acceptance.

I have two very minor comments.

1) While provided in the form of raw excel sheets, the dataset are not fully self-explicit. I recommend that the authors provide unambiguous description and explanation for each variable name, units or coding. Examples of such "readme.txt" file can be found on Dryad website. These files are enormously helpful to improve future re-analysis of the data.

2) Incidentally, I think the reviewers generally warrant acknowledgement, considering their voluntary editorial work.

--

Dr François-Xavier Dechaume-Moncharmont
University of Burgundy, Dijon, France
fx.dechaume@u-bourgogne.fr

Decision letter (RSOS-190587.R1)

22-Aug-2019

Dear Ms Riley:

On behalf of the Editors, I am pleased to inform you that your Manuscript RSOS-190587.R1 entitled "Coping with strangers: how familiarity and active interactions shape group coordination in *Corydoras aeneus*" has been accepted for publication in Royal Society Open Science subject to minor revision in accordance with the referee suggestions. Please find the referees' comments at the end of this email.

The reviewers and Subject Editor have recommended publication, but also suggest some minor revisions to your manuscript. Therefore, I invite you to respond to the comments and revise your manuscript.

- Ethics statement

- Data accessibility

It is a condition of publication that all supporting data are made available either as supplementary information or preferably in a suitable permanent repository. The data

accessibility section should state where the article's supporting data can be accessed. This section should also include details, where possible of where to access other relevant research materials such as statistical tools, protocols, software etc can be accessed. If the data has been deposited in an external repository this section should list the database, accession number and link to the DOI for all data from the article that has been made publicly available. Data sets that have been deposited in an external repository and have a DOI should also be appropriately cited in the manuscript and included in the reference list.

If you wish to submit your supporting data or code to Dryad (<http://datadryad.org/>), or modify your current submission to dryad, please use the following link:
<http://datadryad.org/submit?journalID=RSOS&manu=RSOS-190587.R1>

- **Competing interests**

- **Authors' contributions**

- **Acknowledgements**

- **Funding statement**

Because the schedule for publication is very tight, it is a condition of publication that you submit the revised version of your manuscript before 31-Aug-2019. Please note that the revision deadline will expire at 00.00am on this date. If you do not think you will be able to meet this date please let me know immediately.

To revise your manuscript, log into <https://mc.manuscriptcentral.com/rsos> and enter your Author Centre, where you will find your manuscript title listed under "Manuscripts with Decisions". Under "Actions," click on "Create a Revision." You will be unable to make your

revisions on the originally submitted version of the manuscript. Instead, revise your manuscript and upload a new version through your Author Centre.

on behalf of Kevin Padian (Subject Editor)
openscience@royalsociety.org

Reviewer comments to Author:
Reviewer: 2

Comments to the Author(s)
I carefully read through the new version of the MS and the point-by-point response letter. The authors have made appreciated effort to convincingly address all the comments made by the

referees. The MS is now much clearer. The introduction, materials and methods, and results sections have been rewritten, rearranged and occasionally expanded. I also greatly appreciated the new thorough statistical analyses and the clarity of the methodological description. Despite the fact that I still regret the absence of information about repeatability or inter-individual consistency of the described behaviours, the conclusions of the study are well supported by the neat dataset. I now recommend acceptance.

I have two very minor comments.

1) While provided in the form of raw excel sheets, the dataset are not fully self-explicit. I recommend that the authors provide unambiguous description and explanation for each variable name, units or coding. Examples of such "readme.txt" file can be found on Dryad website. These files are enormously helpful to improve future re-analysis of the data.

2) Incidentally, I think the reviewers generally warrant acknowledgement, considering their voluntary editorial work.

--

Dr François-Xavier Dechaume-Moncharmont
University of Burgundy, Dijon, France
fx.dechaume@u-bourgogne.fr

Reviewer: 1

Comments to the Author(s)

The authors have addressed all my previous concerns and I am happy to recommend the manuscript for publication without further changes.

Author's Response to Decision Letter for (RSOS-190587.R1)

See Appendix B.

Decision letter (RSOS-190587.R2)

30-Aug-2019

Dear Ms Riley,

I am pleased to inform you that your manuscript entitled "Coping with strangers: how familiarity and active interactions shape group coordination in *Corydoras aeneus*" is now accepted for publication in Royal Society Open Science.

You can expect to receive a proof of your article in the near future. Please contact the editorial office (openscience_proofs@royalsociety.org and openscience@royalsociety.org) to let us know if you are likely to be away from e-mail contact -- if you are going to be away, please nominate a co-

author (if available) to manage the proofing process, and ensure they are copied into your email to the journal.

Kind regards,

on behalf of Professor Kevin Padian (Subject Editor)
openscience@royalsociety.org

Appendix A

03-Jul-2019

Dear Ms Riley,

The editors assigned to your paper ("Coping with strangers: how familiarity and active interactions shape group coordination in *Corydoras aeneus*") have now received comments from reviewers. We would like you to revise your paper in accordance with the referee and Associate Editor suggestions which can be found below (not including confidential reports to the Editor). Please note this decision does not guarantee eventual acceptance.

Please submit a copy of your revised paper before 26-Jul-2019. Please note that the revision deadline will expire at 00.00am on this date. If we do not hear from you within this time then it will be assumed that the paper has been withdrawn. In exceptional circumstances, extensions may be possible if agreed with the Editorial Office in advance. We do not allow multiple rounds of revision so we urge you to make every effort to fully address all of the comments at this stage. If deemed necessary by the Editors, your manuscript will be sent back to one or more of the original reviewers for assessment. If the original reviewers are not available, we may invite new reviewers.

- Data accessibility

<http://datadryad.org/submit?journalID=RSOS&manu=RSOS-190587>

- **Competing interests**

- **Authors' contributions**

- **Acknowledgements**

- **Funding statement**

on behalf of Kevin Padian (Subject Editor)
openscience@royalsociety.org

Associate Editor's comments:

Two reviewers have provided commentary on your work, each recommending a range of changes, some that should be a quick fix, others are more substantial and will require further reviewer oversight when you submit your revised paper. Please ensure that you provide a revised manuscript (ideally identifying changes clearly with some format of tracked changes) and a point-by-point response to the comments. Note that in general, only one round of revision is permitted, so we urge you to work hard to fully address the referees' concerns.

Subject Editor Comments to Author:

Thank you for your submission. As you'll see the reviewers were generally positive but there are some issues that need to be addressed. We look forward to your revision.

Many thanks for your continued consideration of our manuscript. We would like to thank both reviewers for their insightful comments. As you will see, we have taken all of them onboard (our responses in italics) and believe we have improved its robustness and readability as a result.

Comments to Author:

Reviewers' Comments to Author:
Reviewer: 1

Comments to the Author(s)

The manuscript "Coping with strangers: how familiarity and active interactions shape group coordination in *Corydoras aeneus*" investigates whether tactile interactions facilitate coordination in pairs and triplets of familiar and unfamiliar

fish. In general, I found the manuscript to be a sound and interesting piece of work on an understudied aspect of social interactions in groups (namely tactile interactions). I have relatively minor comments, however, I would like to the authors to address these before I can recommend publication.

Thank you so much for your helpful comments and appreciation of this work- we agree that our exploration of tactile social interactions is the greatest strength of our study. Your feedback has been very helpful for improving this manuscript, and we appreciate your time and thoughtfulness.

1) Line 145: Please confirm whether it 26, and not 27, pairs that were analysed (to mirror the number of individuals that were tested).

We apologize for this error. We analysed 26 pairs (15 familiar and 11 unfamiliar) for a total of 52 individuals; we have amended this in the text (L160).

2) Line 141 & 161: Please confirm whether the same fish were used in the pairs study as were used in the triplet study.

We did use fish from the same stock population for both the pair and triplet studies; we have clarified this in the text (L143-144).

3) Line 212: Details of the appendix should be referred to that highlight how the authors defined initiator and receiver.

Thank you for this helpful suggestion- we have added information from the appendix to this section and believe this has clarified the manuscript (L219-222).

4) Line 235: Please confirm how you tested for normality following transformation of these data.

Thank you for pointing this out- it was helpful for improving our statistical methodology. We have now completed a nonparametric statistical test (a Spearman rank correlation) on these non-transformed data (L250), which yields qualitatively unchanged results (L299-300) .

For all transformed data, we evaluated normality visually, which we now indicate in the text (L255, L262), and results were qualitatively unchanged when nonparametric tests were performed on non-transformed data (L256-257, L262-263, see appendix).

5) Line 251: I do not think that a quadratic function is an appropriate model to fit to the data here. There is no evidence that as the total number of nudges increases, the proportion of the time together decreases. Have the authors tried to fit a logistic regression to these data? This would satisfy the binomial nature of the data (the fish are together or not) and second this limits the predicted

proportions between 0 and 1. This would be a more suitable fit than the quadratic curve, and would not affect interpretation of results.

Thank you for this insightful comment; we agree that this statistical methodology is not ideal and have addressed this in two ways. The suggestion here (a logistic regression- we initially used a GLM with a quasibinomial distribution. The proportion of time the unfamiliar fish spent together with its group-mates was the response variable, and the number of nudges the unfamiliar fish initiated was the explanatory variable) worked well, and the results are robust and qualitatively consistent with our previous analysis ($F=35.73$, $p<0.001$).

Our only concern with this approach was that, because of the nature of our data, time steps were not truly independent from one another, and treating cohesion as a binomial variable based on time intervals (in our case, seconds) may be problematic. For this reason, we elected to use a one-inflated beta regression model (L268-270), which seems to address our proportion data without making assumptions about independence of time steps. These results are also qualitatively consistent with our previous analysis (L347-348; one-inflated beta regression, $t_{15} = 3.96$, $p < 0.01$, Fig. 5)

6) Line 259: t_{24} - 24 should be subscript.

Thank you for pointing this out- we have corrected this in the manuscript (L281).

7) Line 261: It would be useful to report the % of nudges from the front fish and back fish as percentages out of the 47.3% of nudges that initiated coordinated movement. i.e. they will both be approximately 50% (and not ~25%).

Thank you for pointing this out- we have changed this in the manuscript and appreciate this opportunity to clarify our results. (L286-288).

8) Figure 1: In the text (line 275) the authors report that these data are from both familiar and unfamiliar fish pairs, whereas in the figure the authors report this is data from just the unfamiliar pairs. Please confirm.

Thank you for pointing out this error- we have corrected the figure and results section to reflect that both familiar and unfamiliar pairs were included in this analysis (L297, L303).

Reviewer: 2

Comments to the Author(s)

This manuscript is a descriptive study about a novel behaviour possibly involved

in coordinated movement in a fish species. The MS is very clearly written in a direct and engaging style. I also greatly appreciated the clarity of the methodological description in the Material and Methods section and the important Appendix (would it be possible to move some of this material in the main text?). Most, but not all, conclusions of the study are well supported by the dataset. My overall impression about this manuscript is positive. Yet, some points deserve clarification prior final acceptance.

Thank you very much for your insightful comments and feedback- they have been very helpful, and we believe the manuscript is much improved as a result. We have also included more material from the Appendix in the main text (see our response to Reviewer 1, Comment 3). Our responses to your specific comments appear below.

Major comments.

9) My main criticism is related to the absence of direct investigation of the causal relationship between nudging rate and the degree of coordination between fish. The authors only provide very indirect evidences of this relationship. Strong statements are made in the discussion: "unfamiliar pairs had to engage in a significantly higher nudging rate in order to achieve the same degree of coordination as familiar pairs" (Lines 353-354), "nudging can play a role in aiding coordination between individuals" (line 369). Yet, the authors do not really provide data about the quality or the degree of coordination between fish. To confidently claim that this nudges are involved in the coordination between fish, the authors must first address several issues:

(1) It is crucial to quantify the coordination between fish and characterize the quality of this coordination. There are many metrics of coordination in fish school to be found in the vast literature on collective movement.

(2) Then, the authors must correlate the degree of coordination with the nudging rates.

Thank you for this opportunity to clarify our methodology and more accurately interpret our results. Our study was observational in nature and sought to establish the relationship between nudging, coordination, and familiarity. In order to establish this connection, we examined two main indices of coordination: the amount of time spent in leader/follower dynamics (which we have designated front fish/back fish for this study to interpret as objectively as possible), which was only possible for pairs, and the degree of social cohesion.

The analysis of leader/follower dynamics is a very commonly used metric of coordination in pairs and group settings (Couzin, 2005, King 2009, Jolles 2017, etc), and we chose it for this study because periods of leader/followership represent unambiguous examples of pair coordination. While some studies have made use of automated tracking software to precisely quantify aspects of coordination (including movement tracking of leader-fronted movements),

automated tracking was problematic for our study system due to the Bronze Cory catfish's cryptic colouration; in its natural habitat, the Bronze Cory catfish lives in sandy-bottomed streams in which it is camouflaged (Sands, 1986). Furthermore, the individuals in our study tended to exhibit stress behaviours when placed on artificially coloured substrates that enhanced contrast. For these reasons, and because we were describing aspects of a novel behaviour, automated tracking was not suitable for our study and we confined our analysis to clear instances of leader/followership dynamics.

Consequently, we showed that the use of nudging during periods of leader/follower coordination (which we have called 'coordinated movements') is significantly higher than when fish are in proximity to one another but not engaged in a coordinated movement. We believe this is a valid comparison of nudging associated with higher quality coordination, as cohesion can be designated as relatively passive (and less active) coordination that is a prerequisite for active leader/follower pair movements. In the text we now discuss this explicitly (L232-238, L241-243).

For triplets, where it was not practical to assess leader/followership dynamics, we used social cohesion to demonstrate 1) that unfamiliar individuals were less socially cohesive than their familiar counterparts, and therefore necessarily less coordinated, and 2) that unfamiliar individuals that nudged more frequently were more likely to maintain social cohesion (beyond the effect of proximity, which is a prerequisite for nudging). This observation, in addition to our observations on coordinated movements in pairs, seem to suggest that nudging may be involved in coordination. We do agree that a definitively causal interpretation of nudging is beyond the scope of this observational study, and have amended the manuscript (in particular, the abstract and discussion) to reflect this (L32, L38, L44, L381-382, L397).

10) Were the individually marked in order to be recognized? Firstly, it is a crucial methodological question: how did they recognized each fish (familiar vs non-familiar from a triplet of fish) in the 30-minutes video recordings? The MS is unclear on that matter. Secondly, and maybe more interestingly, ID information allows for the assessment of inter-individual differences in the initiation of these nudges. Are some fish more prone to nudge? Are their nudging rate repeatable?

Thank you for pointing this out. We have amended the text to reflect that, for pairs and triplets, we were able to individually recognize fish by comparing them with their group-mates in size (i.e. body length and width) and coloration. We then noted the characteristics of each individual so that we could distinguish both partners in pairs and all three individuals in triplets (i.e. the unfamiliar fish and each familiar fish) We state this now explicitly (L193-197).

Because we were able to recognize individuals in groups visually, we did not tag or mark the fish. For this reason, we cannot compare the repeatability of

individual differences, although this would be a fascinating subject for future studies.

Minor comments.

11) Lines 164 and followings. About the fish used in the second experiment (based on triplet), were they different fish from experiment 1, or fish randomly sampled from the same stock?

These are fish randomly sampled from the same stock (see our response to Reviewer 1 Comment 2). We have clarified this in the text (L143-144)

12) Lines 190-201. These sentences seem exogenous, here. They break the flow of the paragraph, which deals with the "intentionality" or the "accidental nature" of the nudges. Could it be possible to move these sentences earlier, maybe in the introduction section?

We have moved this section to earlier in the manuscript, specifically in our description of our study species (L124-136). This helps establish that we would not expect these (or any) fish to collide with one another incidentally.

13) Line 252. Provide complete reference (R software foundation 2018).

We have added the complete reference (L524-525).

14) Line 257 and followings. While not incorrect, the presentation of the results is confusing. Earlier in the MS (lines 213-214), the authors claimed that "[they] We present analysis performed on nudges that are initiated with the FRONT PART of the initiator's body". Yet, in this Result paragraph, they systematically presented stats about the nudges initiated by the TAIL. In addition, I do not understand how is it possible that "coordinated movements began with a nudge from the back fish." What does that mean? The fish that was touched led the collective movement?

Thank you pointing this out; we have rearranged this section of the results and believe that this improves its clarity (L277-289). We have also clarified that coordinated movements that began with a nudge from the back fish does indeed mean that the recipient of the nudge led the collective movement (L288-289).

15) Fig. 1. "trasf" is an unusual abbreviation. In addition, the legend is not self-explicit. The reader must be informed that the nudging rate corresponds not to plain number of nudges per unit of time, but to a number of nudges per unit of time spent in proximity. This is not a minor detail because this metric is correct for the lower spatial proximity between unfamiliar fish.

We agree that this figure should be clarified. We have changed the axis label to read 'proportion of time in front' (see also our response to Reviewer 1, Comment 4; we have presented nonparametric analysis of this data using the untransformed proportion of time in front) and added detail to the legend to ensure that the measurement units are clear (L302-306)

16) Lines 320 and following. The quadratic model is a very bad modelling choice, here. By nature, the proportion of time is bounded between 0 and 1. In addition, it is unsurprising that the proportion of time spend together mechanically increases with the absolute number of nudges (and vice versa) up to a point where the number of nudges is so large that the two fish literally stick together. The number of nudge cannot decrease, as suggested by the quadratic relationship.

Thank you for pointing this out. We have reanalysed this relationship using a one-inflated beta regression model (see also our response to Reviewer 1, Comment 5)

--

Dr François-Xavier Dechaume-Moncharmont
University of Burgundy, Dijon, France
fx.dechaume@u-bourgogne.fr

Journal Name: Royal Society Open Science
Journal Code: RSOS
Online ISSN: 2054-5703
Journal Admin Email: openscience@royalsociety.org
Journal Editor: Andrew Dunn
Journal Editor Email: openscience@royalsociety.org
MS Reference Number: RSOS-190587
Article Status: SUBMITTED
MS Dryad ID: RSOS-190587
MS Title: Coping with strangers: how familiarity and active interactions shape group coordination in *Corydoras aeneus*
MS Authors: Riley, Riva; Gillie, Elizabeth ; Horswill, Catharine; Johnstone, Rufus; Boogert, Neeltje; Manica, Andrea
Contact Author: Riva Riley
Contact Author Email: rjr58@cam.ac.uk, rjr58@cam.ac.uk
Contact Author Address 1: Downing Street
Contact Author Address 2:
Contact Author Address 3:
Contact Author City: Cambridge
Contact Author State:
Contact Author Country: United Kingdom of Great Britain and Northern Ireland
Contact Author ZIP/Postal Code: CB2 3EJ
Keywords: sociality, social coordination, tactile interaction, *Corydoras*

Abstract: Social groups comprised of familiar individuals exhibit better coordination than unfamiliar groups, however, the ways familiarity contributes to coordination are poorly understood. Prior social experience likely allows individuals to learn the tendencies of familiar group-mates and respond accordingly. Without prior experience, individuals would benefit from strategies for enhancing coordination with unfamiliar others. We used a social catfish, *Corydoras aeneus*, that utilizes discrete, observable tactile interactions to assess whether active interactions might facilitate coordination, and how their role might be mediated by familiarity. We describe this previously unrecognized physical interaction, “nudges”, and show it to be associated with group coordination and cohesion. Furthermore, we investigated nudging and coordination in familiar/unfamiliar pairs. In all pairs we found that nudging rates were higher during coordinated movements than when fish were together but not coordinating. We observed no familiarity-based difference in coordination or cohesion. Instead, unfamiliar pairs exhibited significantly higher nudging rates, suggesting that unfamiliar pairs compensated for unfamiliarity through increased nudging. In contrast, familiar individuals coordinated with comparatively little nudging. Second, we analysed nudging and cohesion within triplets of two familiar and one unfamiliar individual (where familiar individuals had a choice of partner). Although all individuals nudged at similar rates, the unfamiliar group-mate was less cohesive than its familiar group-mates and spent more time alone. Unfamiliar individuals that nudged their group-mates more frequently exhibited higher cohesion, indicating that nudging facilitated cohesion for the unfamiliar group-mate. Overall, our results suggest that nudges can mitigate unfamiliarity, but that their usage is reduced in the case of familiar individuals, implying a cost is associated with the behaviour.

EndDryadContent

Appendix B

Dear Alice Power and Kevin Padian,

Thank you very much for your further consideration of our manuscript; we are delighted that it has been accepted and are very grateful to the editorial team and both reviewers for your input.

We have addressed the remaining points below (our response in italics) and have indicated where we have made the suggested changes and additions.

Sincerely,
Riva Riley (on behalf of all authors)

Reviewer comments to Author:

Reviewer: 2

Comments to the Author(s)

I carefully read through the new version of the MS and the point-by-point response letter. The authors have made appreciated effort to convincingly address all the comments made by the referees. The MS is now much clearer. The introduction, materials and methods, and results sections have been rewritten, rearranged and occasionally expanded. I also greatly appreciated the new thorough statistical analyses and the clarity of the methodological description. Despite the fact that I still regret the absence of information about repeatability or inter-individual consistency of the described behaviours, the conclusions of the study are well supported by the neat dataset. I now recommend acceptance.

I have two very minor comments.

1) While provided in the form of raw excel sheets, the dataset are not fully self-explicit. I recommend that the authors provide unambiguous description and explanation for each variable name, units or coding. Examples of such “readme.txt” file can be found on Dryad website. These files are enormously helpful to improve future re-analysis of the data.

Thank you for this opportunity to clarify our raw data; we now include a descriptive spreadsheet that clearly and thoroughly defines each variable that was involved in our analysis. We have organized the spreadsheet in the same manner as the raw data, that is, by pairs and triplets. We have also ensured that all variable names and definitions are consistent. Both our raw data and the descriptive spreadsheet included in the electronic supplementary materials.

2) Incidentally, I think the reviewers generally warrant acknowledgement, considering their voluntary editorial work.

We are certainly grateful for our reviewers' insightful input and have taken this chance to acknowledge their time and effort. We have added Dr. Dechaume-Moncharmont and our anonymous reviewer to our acknowledgements (L572-573).

--

Dr François-Xavier Dechaume-Moncharmont
University of Burgundy, Dijon, France
fx.dechaume@u-bourgogne.fr

Reviewer: 1

Comments to the Author(s)

The authors have addressed all my previous concerns and I am happy to recommend the manuscript for publication without further changes.

Thank you again for your time, effort, and insight; your thoughtful comments have led to marked improvements on this manuscript.